

# Deglacial climate changes as forced by ice sheet reconstructions

Nathaelle Bouttes[1], Fanny Lhardy[1,2], Aurélien Quiquet[1,3], Didier Paillard[1], Hugues Goosse[4], Didier M. Roche[5]

[1]Laboratoire des Sciences du Climat et de l'Environnement, LSCE/IPSL, CEA- CNRS- UVSQ, Université Paris- Saclay, Gif- sur- Yvette, France
[2]Max Planck Institute for Meteorology, Hamburg, Germany
[3]Univ. Grenoble Alpes, CNRS, IRD, Grenoble INP, IGE, 38000 Grenoble, France
[4]Earth and Life Institute, Université catholique de Louvain, Louvain-la-Neuve, Belgium.
[5]Earth and Climate Cluster, Faculty of Science, Vrije Universiteit Amsterdam, Amsterdam, The Netherlands

*Correspondence to*: Nathaelle Bouttes (Nathaelle.bouttes@lsce.ipsl.fr)

**Abstract.** During the Last Deglaciation, the climate evolves from a cold state at the Last Glacial Maximum at 21 ka with large ice sheets, to the warm Holocene at ~9 ka with reduced ice sheets. The deglacial ice sheet melt can impact the climate through multiple ways: changes of topography and albedo, bathymetry and coastlines, as well as fresh water fluxes. In the PMIP4 protocol for deglacial simulations, these changes can be accounted or not depending on the modelling group choices. In addition, two ice sheet reconstructions are available (ICE-6G_C and GLAC-1D). In this study, we evaluate all these effects related to ice sheet changes on the climate using the iLOVECLIM model of intermediate complexity. We show that the two reconstructions yield the same warming to a first order, but with a different amplitude (3.9°C with ICE-6G_C and 3.8°C with GLAC-1D) and evolution. We obtain a stalling of temperature rise during the Antarctic Cold Reversal (from ~14 ka to ~12 ka) similar to proxy data only with the GLAC-1D ice sheet reconstruction. Accounting for changes in bathymetry in the simulations results in a cooling due to a larger sea ice extent and higher surface albedo. Finally, fresh water fluxes result in AMOC drawdown, but the timing in the simulations disagrees with proxy data of ocean circulation changes. This questions the links between reconstructed fresh water fluxes from ice sheet melt and recorded AMOC weakening and their representation in models.

## 1 Introduction

The Last Deglaciation is a time of large climate transition from the cold Last Glacial Maximum (LGM) with large ice sheets at ~21 thousand years ago (ka), to the warmer Holocene with reduced ice sheets at ~9 ka (Clarke et al., 2012). During that time, the global mean temperature likely rises by 4.5±0.9°C (Annan et al., 2022), and possibly 6.1±0.4°C (Tierney et al., 2020).



Deglacial changes are neither smooth nor uniform. Rapid regional changes occur at different times depending on location. In particular, the Northern and Southern high latitudes display a markedly different behaviour. The temperature rise in the Southern Hemisphere starts earlier than in the Northern Hemisphere on average (Shakun et al., 2012). In Antarctica, the

warming then halts between ~14.8 ka and 13 ka at the time of the Antarctic cold reversal (ACR, Jouzel et al., 2007). In Greenland, the temperature abruptly increases at ~14.7 ka at the beginning of the Bolling-Allerod, then drops from 12.8 ka to 11.6 ka during the Younger Dryas (Buizert et al., 2014). As the ice sheets shrink, especially in the Northern Hemisphere, large fluxes of fresh water go to the ocean and lead to a considerable increase of sea level by around 120-130 m (Lambeck et al., 2014; Gowan et al., 2021). Besides temperature, proxy records indicate large changes of ocean circulation during the transition

(McManus et al., 2004; Ng et al., 2018). They suggest a strong weakening of the Atlantic Meridional Overturning Circulation (AMOC) at the beginning of the deglaciation from ~19 ka followed by a resumption at ~15 ka. Ocean circulation changes result in a modification of heat transport and variations of temperature in the Northern and Southern Hemispheres. Sea ice changes are also likely to play a crucial role in oceanic circulation changes, because convection and deep-water formation are linked to sea ice formation. The sea ice formation rejects salt and thus makes surrounding waters saltier and denser. However,

the links between ice sheets, ocean circulation and inter-hemispheric temperature evolution during the transition are difficult to disentangle.

To better understand the drivers of the deglaciation and the response of the different components of the Earth system, modelling groups have endeavoured to run transient simulations from the Last Glacial Maximum to the Holocene. To do so, models need to account for changes in orbital forcing greenhouse gases and ice sheets. Previous studies have focused on the evolution of

climate in simulations with prescribed ice sheets from reconstructions (e. g. Menviel et al., 2011; He et al., 2013; Kapsch et al., 2022) or with interactive ice sheets (e. g. Bonelli et al., 2009; Ganopolski and Calov, 2010; Quiquet et al., 2021), which is technically more challenging. The warming and ice sheet melt are largely driven by the insolation change due to the evolution of orbital parameters, especially the insolation increase in the high Northern latitudes in summer (Berger et al., 1978) as well as the atmospheric $CO_2$ concentration increase (Bereiter et al., 2015; Gregoire et al., 2015).

The Last Deglaciation is an ideal test bed to evaluate models during a period of large warming and to better understand the underlying processes, as many proxy data exist for that period. As such, more and more models have undergone simulating





the transition and a new PMIP4 protocol has been set up for the Last Deglaciation (Ivanovic et al., 2016) to facilitate the

comparison of model results in a common framework. In the protocol, the main prescribed boundary conditions are orbital

parameters, atmospheric greenhouse gases and ice sheets. While the insolation and greenhouse gas concentration evolution are

relatively well known, the ice sheet evolution is more uncertain, hence two different reconstructions are included in the PMIP4

protocol: ICE-6G_C and GLAC-1D (Figure 1, Argus et al., 2014; Peltier et al., 2015; Briggs et al., 2014; Tarasov et al., 2012;

Tarasov and Peltier 2002; Gowan et al., 2021). These reconstructions are obtained from inverse modelling based on GPS data

and local sea level records. Large uncertainties remain, mostly due to the viscosity model use for solid Earth.

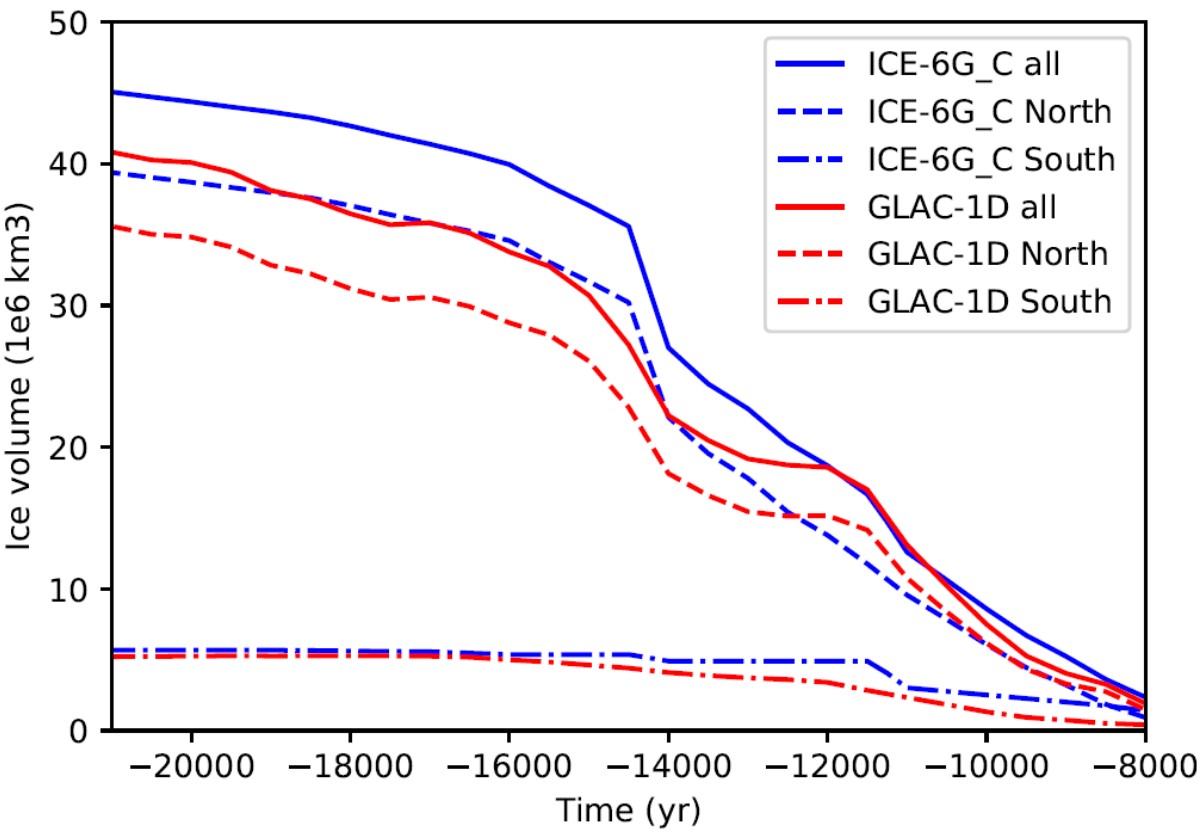


**Figure 1: Global ice sheet volume evolution ($10^6$ km³) for the two reconstructions (solid lines), with contributions of Northern (dashed lines) and Southern (dash-dotted lines) ice sheets (data from Ivanovic et al., 2016).**

Besides the ice sheet reconstruction itself, the PMIP4 protocol lets to some extent modelling group decide how to implement

ice sheet related changes such as land-sea mask (i.e. coastlines), bathymetry and fresh water fluxes. The purpose of this

flexibility in the protocol is to allow any willing modelling group to join the effort, whatever the complexity of their model,

from intermediate complexity to state-of-the art General Circulation Models (GCMs). However, not all models are able to run

the entire range of possible simulations, it is thus crucial to evaluate the impact on climate of the different choices in setting

up the experiments. More specifically, model groups have three choices related to ice sheets: (a) the choice of ice sheet

reconstruction: ICE-6G_C or GLAC-1D; (b) to only account for topography/ice mask (i.e. albedo changes), or also land-sea

mask and bathymetry changes and (c) to account or not for the flux of meltwater from ice sheets to the ocean, and if included

to do so in a uniform way or a more realistic way such as by routing melted water towards the ocean. The relative effect of all

these individual modelling choices on the transient climate simulated is unclear.


The effect of ice sheet related changes has been analysed to some extent in past studies, but not all of them, usually not within

the PMIP4 deglacial framework, and not in a systematic way. For example, an evolving bathymetry (including changes of

coastlines) has recently been implemented in a GCM (Meccia and Mikolajewicz, 2018), and while it has been used in transient

runs of the Last Deglaciation (Kapsch et al., 2022), the impact of evolving bathymetry compared to fixed bathymetry on

climate has not been evaluated yet. Concerning fresh water fluxes, they were usually not included in early studies of deglacial

changes, (Timm and Timmermann, 2007; Roche et al., 2011; Smith and Gregory, 2012), possibly explaining the lack of rapid

regional changes in these simulations. In contrast, in simulations with fresh water fluxes, rapid events were observed in addition

to the global deglacial warming (Liu et al., 2009; Menviel et al., 2011; Bethke et al., 2012; He et al., 2013; Obase and Abe-

Ouchi, 2019). However, the simulations that show temperature and circulation changes in best agreement with data were

usually run with tuned fresh-water fluxes (Liu et al., 2009; Menviel et al., 2011; He et al., 2013; Obase and Abe-Ouchi, 2019).

On the one hand, such prescribed fresh water fluxes are usually inconsistent with the volume change inferred from ice sheet

reconstructions. On the other hand, simulations run with the fresh water flux evolution derived from ice sheet reconstructions

show disagreement with observed climate changes (e.g. Bethke et al., 2012; Kapsch et al., 2022).

Here we evaluate the impact of ice sheet led changes, i.e. topography and albedo, bathymetry and coastlines, as well as fresh water fluxes, on climate, especially temperature and ocean circulation. For this, we use iLOVECLIM, a climate model of intermediate complexity, fast enough to run several simulations and test these different effects. This work will help other modelling groups decide which choices to make in terms of ice sheet related changes.

## 2 Method

### 2.1 iLOVECLIM model

We use the iLOVECLIM model, which is a code fork and evolution of the LOVECLIM model (Goosse et al., 2010). iLOVECLIM is an intermediate complexity model which shares the same atmosphere, ocean, sea ice and terrestrial biosphere components as LOVECLIM. The ocean grid has a 3°x3° resolution with 20 irregular vertical levels while the atmosphere is

on a T21 grid with three vertical levels. It can simulate ~1000 years / day, making it very suitable for computing long term climate changes such as the work undertaken here.

To run simulations of the last deglaciation following the PMIP4 protocol, we have added modifications to account for the evolution of the ice sheets as reconstructed in ICE-6G_C and GLAC-1D. We consider the change of ice sheet topography and ice mask, the change of bathymetry and coastlines, as well as the routing of fresh water from ice sheet melt to the ocean.


### 2.2 Modification of ice sheet topography and ice mask

The ice sheet geometry change is accounted for in the atmospheric component of the model through two variables: the surface elevation (topography) and the ice sheet mask (albedo). These two variables are updated interactively in the course of the simulation to follow the ICE-6G_C or GLAC-1D reconstructions, using anomalies with respect to ETOPO1 (NOAA, 2009).

The updates are done abruptly, i.e. without temporal interpolation, every 500 years for the ICE-6G_C reconstruction and every 100 years for the GLAC-1D reconstruction. The ice sheet albedo in the iLOVECLIM model is set to the constant and homogeneous value of 0.85.





**2.3 Modification of bathymetry and coastlines**

We have extended the computation of bathymetry developed in Lhardy et al. (2021) for the deglaciation. Following the same procedure, a new bathymetry has been adapted to the iLOVECLIM ocean grid based on those provided in the ICE-6G_C and GLAC-1D reconstructions at regular intervals starting at 21ka. The bathymetry anomaly (with respect to the pre-industrial) at each time step is added to the pre-industrial bathymetry based on ETOPO1. The bathymetry associated with each ice sheet reconstruction is interpolated into the ocean grid in an automated way, with a few limited manual adjustments in straits or key passages. We decide to maintain passages open at our grid resolution when they were open in the reconstructions. Since the ice sheet reconstruction (and thus bathymetry) is provided every 100 years for the GLAC-1D reconstruction and every 500 years for the ICE-6G_C reconstruction, we keep the same frequency for the bathymetry update in iLOVECLIM. With the reconstructed bathymetry, the ocean volume increases across the deglaciation due to the ice sheet volume decrease (Figure 2). The initial volume of the ocean at the LGM is lower with ICE-6G_C ($1.291 \ 10^{18} \ m^3$) than with GLAC-1D ($1.296 \ 10^{18} \ m^3$) as more ice is stored in the ice sheets reconstructed with ICE-6G_C. Apart from this relatively constant shift, the evolution is rather similar in the two reconstructions until 14.5ka. From around 14.5ka, the ocean volume increase accelerates, and this acceleration is larger with ICE-6G_C so that the shift between both reconstructions is reduced. From around 12ka and as we get nearer to the Pre-Industrial, the ocean volume becomes roughly the same in both reconstructions: as the ice sheet topography is prescribed as an anomaly with respect to ETOPO1 in the model, when the additional ice sheet volume disappears both ice sheet forcings converge.

There is more variability in the GLAC-1D evolution due to the update of bathymetry every 100 years compared to every 500 years for ICE-6G. On top of that, from 12.5 ka the evolution is more variable in GLAC-1D than ICE-6G, with accelerations and slow-downs.





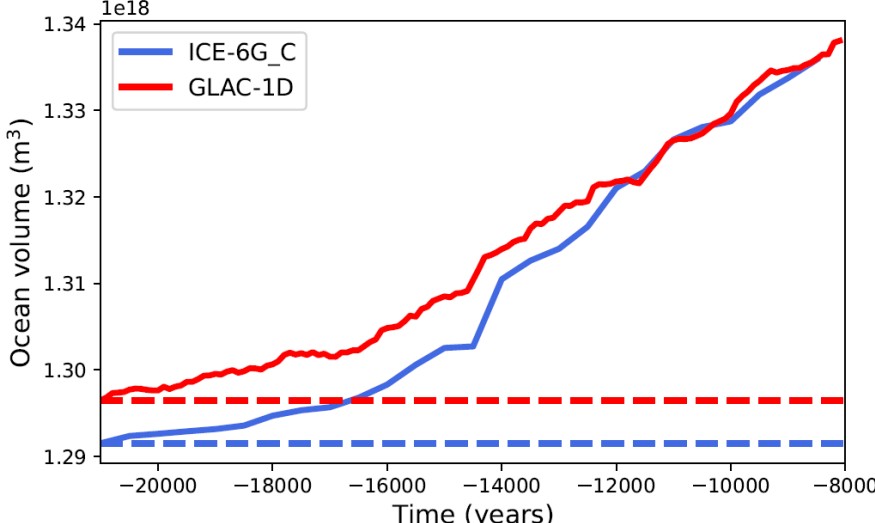


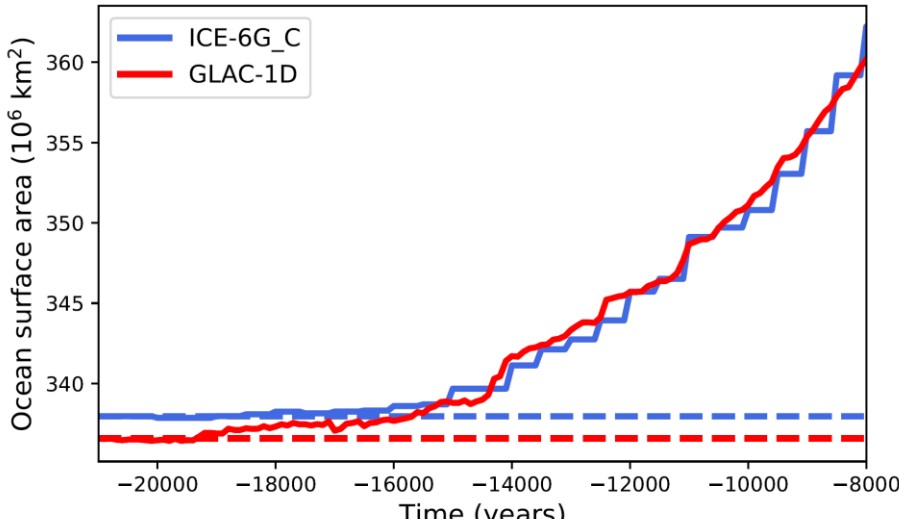

**Figure 2. Evolution of a) the ocean volume (m³) with the ICE-6G_C and GLAC-1D bathymetry reconstructions and b) the surface area (10⁶ km²). The dashed lines indicate the values for the simulations with fixed bathymetry.**


With the evolving bathymetry, we generate evolving land-sea masks (Figure 3). At the LGM, the continental area is larger than at the Pre-Industrial due to the lower sea level (Figure 2b). It then decreases throughout the deglaciation while the sea





level rises and new ocean grids appear. At the LGM, the new land-sea mask implies a closing of the Bering strait and Canadian

archipelago. The strait of Gibraltar is maintained open throughout the simulation.






**Figure 3. Land-sea masks generated for the ICE-6G_C and GLAC-1D reconstructions.**



Every time the bathymetry is updated in the model, conservation of salt is ensured by redistributing the excess (or loss) of salt

in the global ocean. In the new oceanic cells, the variables (such as temperature) are initialised using the mean value of neighbour cells. When new continent grids appear, the vegetation model computes a new albedo given the vegetation distribution computed interactively by the model.

## 2.4 Routing of melted water from ice sheets

In iLOVECLIM, contrary to the original LOVECLIM model, the routing is no longer done in fixed routing basins. Rather, water routing is computed along the greater slope of the model topography and is updated accordingly whenever the topography is updated. This allows for a portable evolution of the first order changes in river routing along the deglaciation. The routing thus computed is used for the precipitation (interactive) and for the anomalous freshwater flux arising from the ice-sheet melt. The latter is computed as the change in ice-sheet thickness between two reconstruction snapshots, averaged as

an annual flux and applied homogeneously over the year. The water fluxes are added as a water flux into the ocean model hence changing the salinity.

## 2.5 Simulations following the PMIP4 protocol

All transient simulations are forced with evolving orbital parameters (Berger et al., 1978), as well as the increased concentration

of greenhouse gases, notably $CO_2$ (Bereiter et al., 2015), $CH_4$ (Loulergue et al., 2008) and $N_2O$ (Schilt et al., 2010), following the PMIP4 deglacial protocol (Ivanovic et al., 2016). To evaluate the impact of ice sheet led changes we have run a set of simulations (Table 1) in which we consider:

- The role of ice sheet elevation and albedo: in the first two simulations, only the topography and albedo changes from ice sheets are accounted for, with the two different prescribed reconstructions: ICE-6G_C and GLAC-1D. In these
simulations, the bathymetry and land-sea mask are fixed to the LGM ones.
- The role of bathymetry: two simulations are run with the bathymetry and land-sea mask changes on top of topography changes for the two reconstructions.



- The role of fresh water fluxes from ice sheet melting: a set of simulations with fresh water fluxes from the melting ice sheets are run for the GLAC-1D reconstruction.

The simulations are started from 5,000 year-long spin-up runs with LGM conditions (GLAC-1D, ICE-6G), consistent with the deglaciation protocol.

| Simulation | ICE-6G_C | GLAC-1D | Bathymetry | FWF |
|---|---|---|---|---|
| ICE-6G-fixed bathy | x | | | |
| ICE-6G-evolving bathy | x | | x | |
| GLAC-1D-fixed bathy | | x | | |
| GLAC-1D-evolving bathy | | x | x | |
| GLAC-1D FWF | | x | x | x |
| GLAC-1D FWF/3 | | x | x | x intensity divided by 3 |
| GLAC-1D FWF/4 | | x | x | x intensity divided by 4 |

**Table 1. Summary of the simulations**



## 3 Results

### 3.1 Impact of different ice sheet reconstructions (topography and albedo)

The effects on climate of the two ice sheet reconstructions (ICE-6G_C and GLAC-1D) are first compared when only considering the changes of topography and albedo (no bathymetry nor fresh water changes). These simulations will serve as a reference for the other ones with the added effects of bathymetry and fresh water fluxes.

In both simulations, the global mean temperature is ~4°C colder at the LGM compared to the pre-industrial (Figure 4), in the range of the PMIP4 models temperature change (3.3° to 7.2°C; Kageyama et al., 2021) and reconstructions (Annan et al., 2022). The global mean temperature is slightly warmer in the simulation with the GLAC-1D reconstruction compared to the simulation with ICE-6G_C due to slightly smaller and lower ice sheets with GLAC-1D (Ivanovic et al., 2018; Figure 1). This temperature difference is observed until ~11.5 ka, when temperatures from both simulations have similar values, due to the ice sheet volume evolution. Indeed, the ice sheet height and extent become similar for both reconstructions from ~11.5 ka. The global mean temperature evolution is relatively similar to proxy data reconstruction (Shakun et al., 2012), with a progressive warming until ~14ka. At this time, while the data show a slow-down and then a decrease in temperature, the temperature with ICE-6G continues to increase while the temperature with GLAC-1D shows a stalling more similar to data, but at a later date. This can be linked to the Northern hemisphere ice sheet reconstructions: in GLAC-1D, following a strong ice volume decrease until 14 ka, the volume stays constant from 14 ka to 12 ka, while it continues to decrease, although at a lower rate, with ICE-6G_C. From ~11 ka, both data and model simulations show an increase in temperature followed by a plateau at the beginning of the Holocene.





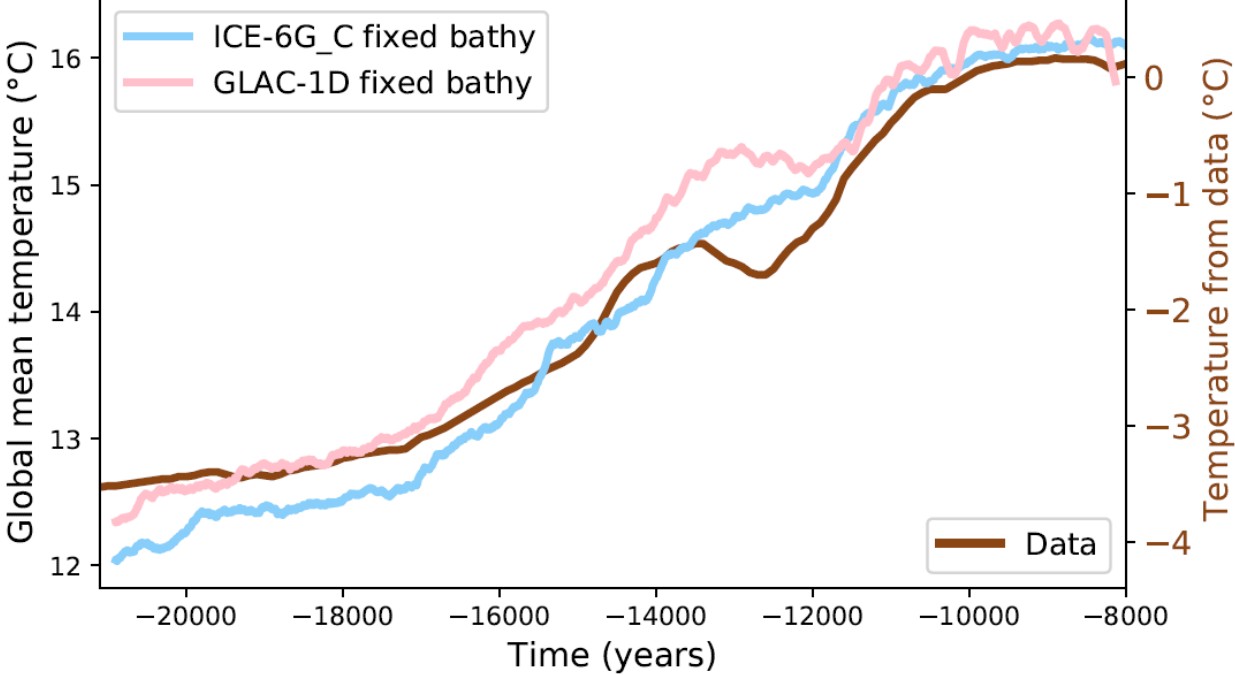

**Figure 4. Global mean annual temperature evolution (°C) for the two simulations with the two ice sheet reconstructions (with fixed**
**bathymetry and no fresh water flux), using a running mean over 100 years. The proxy data, which are shown as anomalies, are from**
**Shakun et al. (2012).**

The sea surface temperature distribution at the LGM and the warming pattern at the beginning of the Holocene at ~8 ka

compared to the LGM are also similar in the two simulations (Figure 5a). Most of the warming at 8ka has occurred in the

North Atlantic and Southern Ocean, where deep water forms and penetrates in the ocean interior.



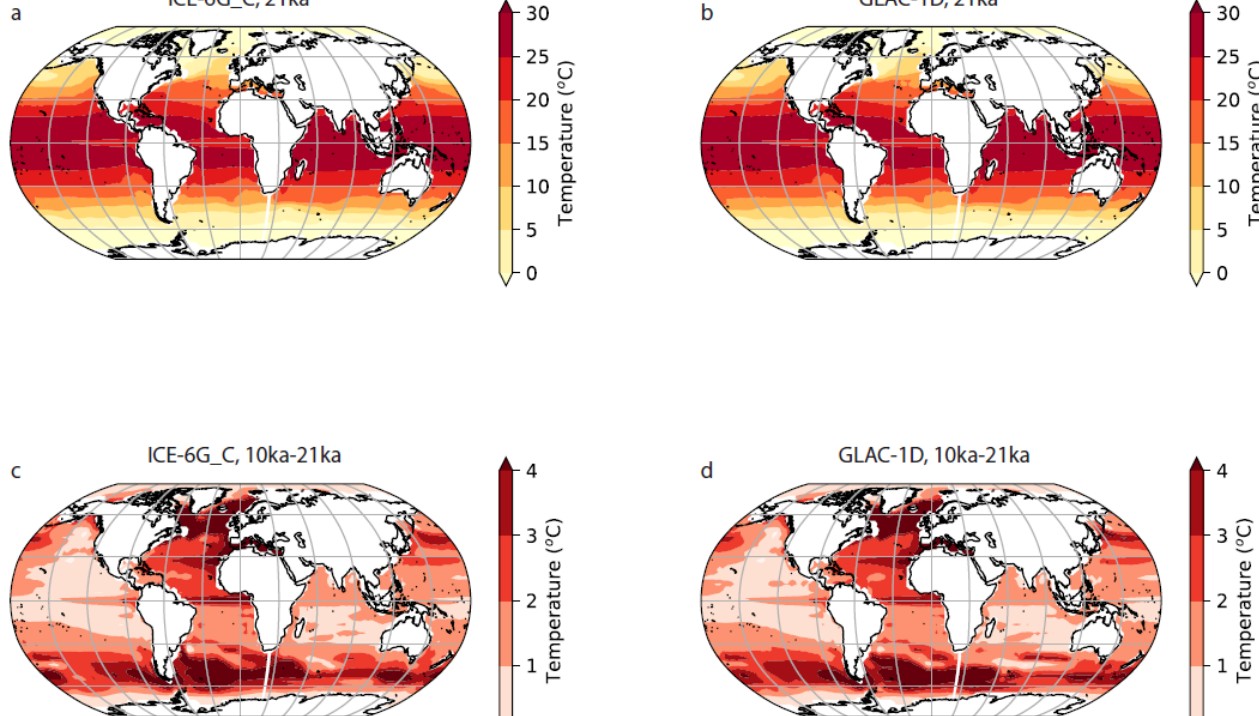

**Figure 5. Ocean sea surface temperature (°C) at (a and b) 21ka and (c and d) difference between 10ka and 21ka. The left panel is with ICE-6G_C, the right panel with GLAC-1D.**


In summary, the first order change of temperature is similar in the two simulations with the two reconstructions, and in relatively good agreement with global mean data. The main differences are a temperature shift from 21 ka to 12ka, and a difference in the evolution between 14 ka and 12 ka, at the time of the Antarctic Cold Reversal and Younger Dryas.

**3.2 Impact of evolving bathymetry and land-sea mask (vs fixed ones)**


On top of the changes of topography and albedo, we have run simulations where we also modify the bathymetry and land-sea mask (later referred to as "with bathymetry") for the two ice sheet reconstructions. The evolving bathymetry and land-sea mask lead to an increase of the ocean volume (Figure 2), and hence a decrease of the global salinity as the global salt content is conserved (Figure 6). The salinity change at the LGM that is computed directly from the volume change (1.4 psu for ICE-




6G_C and 1.3 psu for GLAC-1D)) is larger than the one prescribed in the standard protocol without bathymetry change (1

psu).

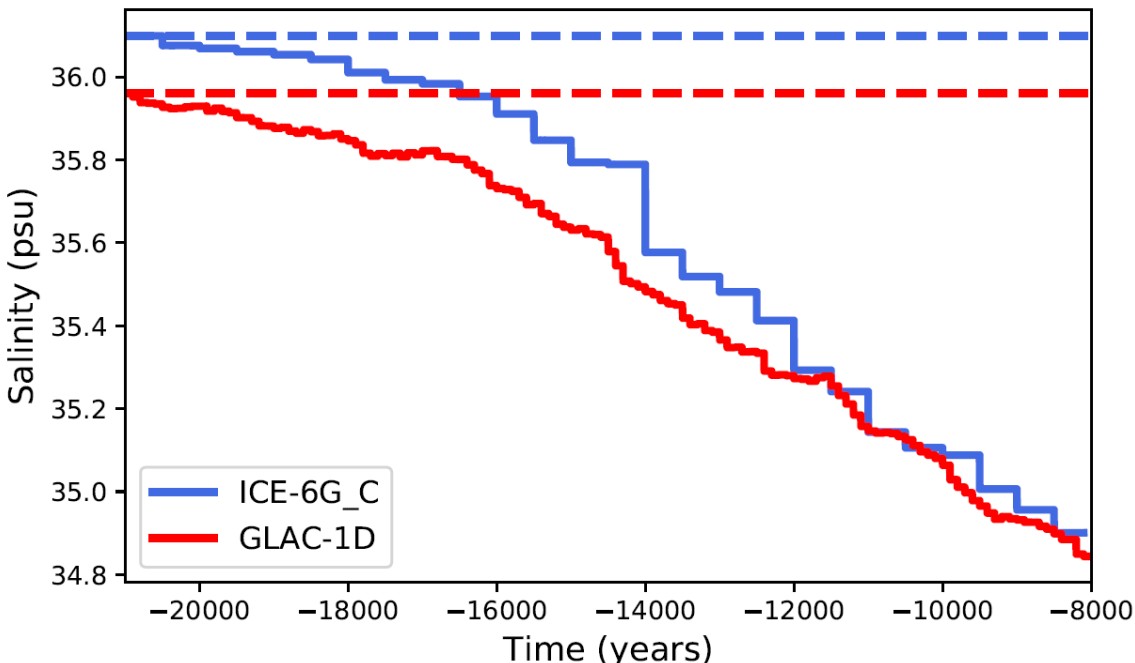

**Figure 6. Evolution of the global mean ocean salinity (psu) simulated in the model with the ICE-6G_C and GLAC-1D**
**reconstructions. The dashed lines indicate the values for the simulations with fixed bathymetry.**

Accounting for the progressive change of bathymetry over time due to the ice sheet changes has a limited impact on the global

mean temperature: its evolution is relatively similar with or without accounting for bathymetry changes (Figure 7a, b) apart

from small changes, such as between 13 ka and 12 ka with ICE-6G, and a constant shift towards slightly colder temperatures

from ~12 ka onwards with ICE-6G_C and from ~14 ka onwards with GLAC-1D.



**Figure 7. Evolution of (a,b) global mean temperature (°C), (c, d) temperature (°C) at NGRIP (Greenland), (e,f) temperature (°C) at EDC (Antarctica). For the simulations, the left panels are with the ICE-6G_C reconstruction, the right ones with GLAC-1D. The simulated results are shown as running mean over 100 years.**

The shift of temperature towards colder values is also visible in the temperature evolution in Greenland (Figure 7c and d) and Antarctica (Figure 7e and f), with a larger amplitude than at global scale. With both reconstructions, this shift is visible mostly from ~14 ka onwards. With ICE-6G_C, Figure 7e also shows a rapid increase then decrease of temperature between 14 ka and 12 ka in Antarctica.



The shift towards colder temperature with evolving bathymetry is mainly due to the albedo change associated with the land-sea mask modification. As shown on Figure 8, the albedo becomes higher in the simulation with evolving bathymetry, a difference that becomes more visible from ~15-14 ka onward, at the same time as the shift in temperature observed on Figure 7. In these simulations, the continental surface area decreases with time while the ocean expands due to the ice sheet volume decrease. While the ocean volume area starts changing earlier, the change in surface area only starts to be significant from ~15 ka (Figure 2). At high latitudes where it is cold enough to have sea ice, the continental surface is replaced by surfaces with sea ice, which has a higher albedo, in particular in summer, leading to the colder temperatures.


**Figure 8. Evolution of surface mean albedo over (a, b) all globe, (c,d) the Northern Hemisphere from 65°N to 90°N, (e,f) the Southern Hemisphere from 90°S to 65°S. The left panels are with the ICE-6G_C reconstruction, the right ones with GLAC-1D.**






In agreement with the albedo change, the sea ice area is also significantly impacted by the evolving bathymetry (Figure 9), because the ocean surface area, especially in the North Atlantic and Arctic, but also in the Southern Ocean, is very different

when accounting for bathymetry changes. Due to the sea level increase, the ocean surface in the high latitudes of the North Hemisphere increases with time, leaving more surface for sea ice to form where it is cold enough. From 15 ka onwards, this leads to more sea ice in the simulation with evolving bathymetry compared to the simulation with fixed bathymetry. With fixed bathymetry, the ocean surface is kept the same as the LGM one throughout the simulation (Figure 2b), and the primary effect governing sea ice changes is the warming resulting in less area covered by sea ice over time. With evolving bathymetry,

this effect is counteracted by the increase of ocean surface at high latitudes so that the sea ice area shows a more limited evolution.





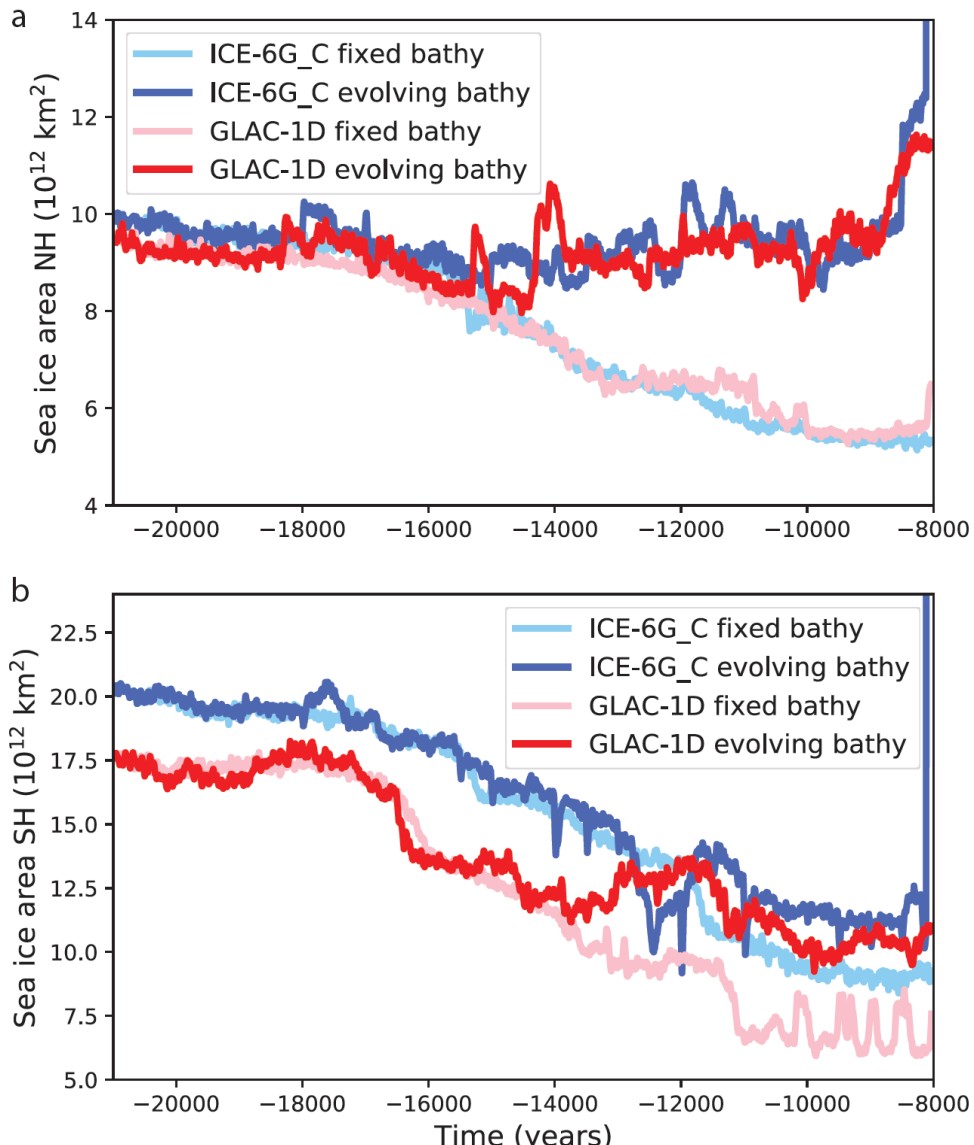

**Figure 9. Evolution of sea ice area ($10^{12}$ km$^2$) in the Northern and Southern Hemispheres.**


In the Southern Ocean, the land-sea mask changes less (Figure 3), and later than in the North Atlantic, yielding apparent

changes between simulations with and without bathymetry at a later date than in the Northern Hemisphere. The difference

between simulations with and without bathymetry starts to be visible from 14 ka with GLAC-1D, and from 13 ka with ICE-

6G. As can be seen on Figure 1, the Antarctic ice sheet starts melting earlier with GLAC-1D compared to ICE-6G, we thus

observe a stronger and earlier change in sea ice with the two GLAC-1D simulations than the ICE-6G simulations. With both

reconstructions, apart from between 13 ka and 12 ka with ICE-6G, having an evolving bathymetry leads to a larger sea ice

area compared to the simulation with fixed bathymetry, as the ocean surface where sea ice can form increases with evolving

bathymetry.

The sea ice formation is tightly linked to convection and AMOC changes: the sea ice forms where it is cold enough for water

to freeze, which corresponds to cold places where the water becomes denser. In the North Atlantic, deep convection occurs

where the heat loss from the ocean to the atmosphere is large enough for the salty and warm water coming from the south to

significantly cool and make the water dense enough to convect. The dominant effect of sea ice is to isolate the ocean from the

atmosphere, preventing heat loss and hence convection, so that convection takes place at the sea ice edge, where it is very cold

and heat loss can take place. In the Southern Ocean, the dominant effect of sea ice is the release of brines (very salty water)

during sea ice formation. This makes the water denser and favours convection.

At the LGM, due to the colder climate, the simulated Northern Hemisphere sea ice edge is shifted to the south with respect to

the pre-industrial with both ice sheet reconstructions (Figure 10a and b), and the mixed layer depth is deep, showing strong

convection in the Iceland, Norwegian and Irminger Basins. At 10 ka, with the warmer conditions the sea ice edge retreats

toward the north and the mixed layer depth is reduced in all simulations (Figure 10). However, the convection sites differ

between simulations. In particular, while there is a convection site in the Labrador Sea in the ICE-6G simulations with fixed

bathymetry, this convection site is not visible anymore in the simulation with evolving bathymetry, where the sea ice edge is

shifted to the south. This is due to the colder conditions in the simulation with evolving bathymetry. The sea ice cover isolates

the ocean from the atmosphere preventing cooling and deep convection from taking place in the Labrador Sea. Instead, another

convection site is observed south of Greenland.





**Figure 10. Winter (DJF) mixed layer depth (m) and sea ice edge (defined by 15% concentration) at different time steps in the fixed bathy and evolving bathy simulations, with the two reconstructions.**

This difference in convection sites observed in the model (Renssen et al., 2005) could explain the strong change in AMOC

strength in ICE-6G_C (Figure 11a): the AMOC strength decreases from 12 ka when accounting for bathymetry change while





it stays relatively constant with a fixed bathymetry, leading to a ~10 Sv difference between the two simulations. This is not the case with GLAC-1D where only small changes between the two simulations can be seen. This can also be seen in the meridional streamfunction shown on Figure 12. With evolving bathymetry and ICE-6G_C, the AMOC is weaker and shallower at 10 ka compared to the simulation with fixed bathymetry (with ICE-6G). In the GLAC-1D simulations, the sea ice edge and

convection sites are also shifted southward in the simulation with evolving bathymetry compared to the one with fixed bathymetry (Figure 10d and f), but the latitude change is smaller, and not sufficient to shut down the Labrador Sea convection site, hence it has a limited effect compared to the ICE-6G_C simulations on the AMOC evolution (Figures 11 and 12).

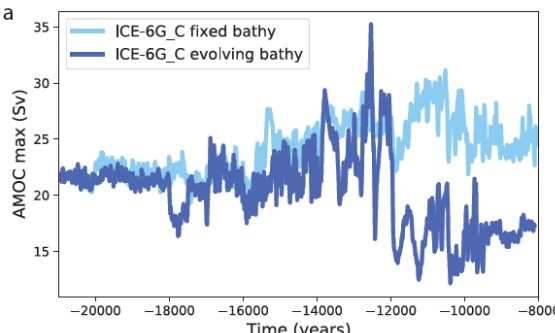
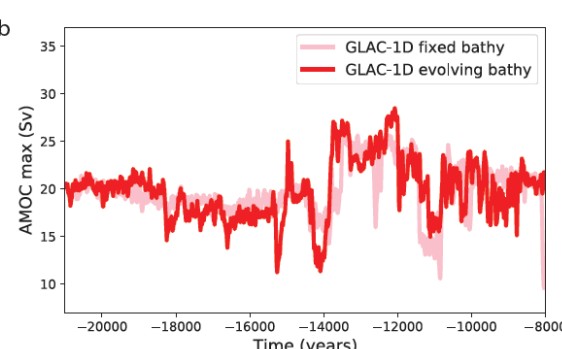

**Figure 11. Evolution of the maximum strength of the Atlantic Meridional Oceanic Circulation (Sv). The left panel is for the ICE-6G_C reconstruction, the right one for GLAC-1D. The simulated ruslts are shown as running mean over 100 years.**






**Figure 12. Meridional overturning streamfunction (Sv) with the two reconstructions at different time steps, with fixed bathymetry and evolving bathymetry. The dark line indicates the limit between the Southern Ocean south of 32° S and the Atlantic Ocean north of 32° S.**



In summary, accounting for bathymetry changes has limited impact on the global mean temperature change, but can lead to large changes in regional climate, such as the convection sites, sea ice, AMOC and temperature in the North Atlantic. This depends on the reconstruction, the model grid resolution, and probably also on the model sensitivity.

## 3.3 Impact of fresh water flux from melting ice sheets

In addition to topography and bathymetry, changes of ice sheets can also have an impact on ocean circulation and climate through the input of fresh water coming from the melting of the ice. We now consider this impact by accounting for fresh water being routed from the site of ice sheet melting to the nearest ocean grid cell. The total water flux from ice sheet melting is relatively small (less than 0.1 Sv) until around 15 ka with both reconstructions (Figure 13a). At 15 ka, the flux becomes very large, up to more than 0.5 Sv, corresponding to meltwater pulse 1A (Deschamps et al., 2012). After 14 ka, the fluxes from the two simulations differ more, with the flux from GLAC-1D showing more variability (due partly to the higher frequency update).





**Figure 13. Total fresh water flux (Sv) to the ocean due to ice sheet melting and comparison with other studies (Menviel et al., 2011; Obase and Abe-Ouchi, 2019). The values for TRACE-21k (Liu et al., 2019) have been obtained from https://www.cgd.ucar.edu/ccr/TraCE/TraCE.22,000.to.1950.overview.v2.htm.**





The fresh water input has a direct effect on convection as it reduces the surface salinity in convection zones, leading to reduced AMOC strength. This is the case with the FWF simulation showing a reduction of AMOC strength starting at around 15 ka (when the fresh water flux becomes very large), and a shut down from around 14 ka (Figure 14). In this simulation, the AMOC does not recover and the simulation crashes at year 11792 BP, i.e. after 9208 years of simulations, for an unknown reason. The

response to a fresh water flux is very model dependent (Gottschalk et al., 2019). Hence, to emulate the response of model less sensitive to fresh water inputs, we have reduced the flux by a factor of 3 and 4. With the reduction of a factor 3 (FWF/3), the AMOC still collapses and recovers only around 8 ka. With a larger factor of 4 (FWF/4), the AMOC is weakened but quickly recovers. Similar changes of AMOC have been observed in an iLOVECLIM version coupled to an ice sheet model (Quiquet et al., 2021). As in other modelling studies using realistic fresh water fluxes computed from ice sheet melt (Kapsch et al.,

2022), the timing of the AMOC weakening disagrees with data (McManus et al., 2004; Ng et al., 2018), taking place too late at around 15 ka when the fresh water flux becomes large, while proxy data indicate an AMOC weakening from ~17 ka, when the fresh water flux is too small to have an impact on the simulated AMOC.



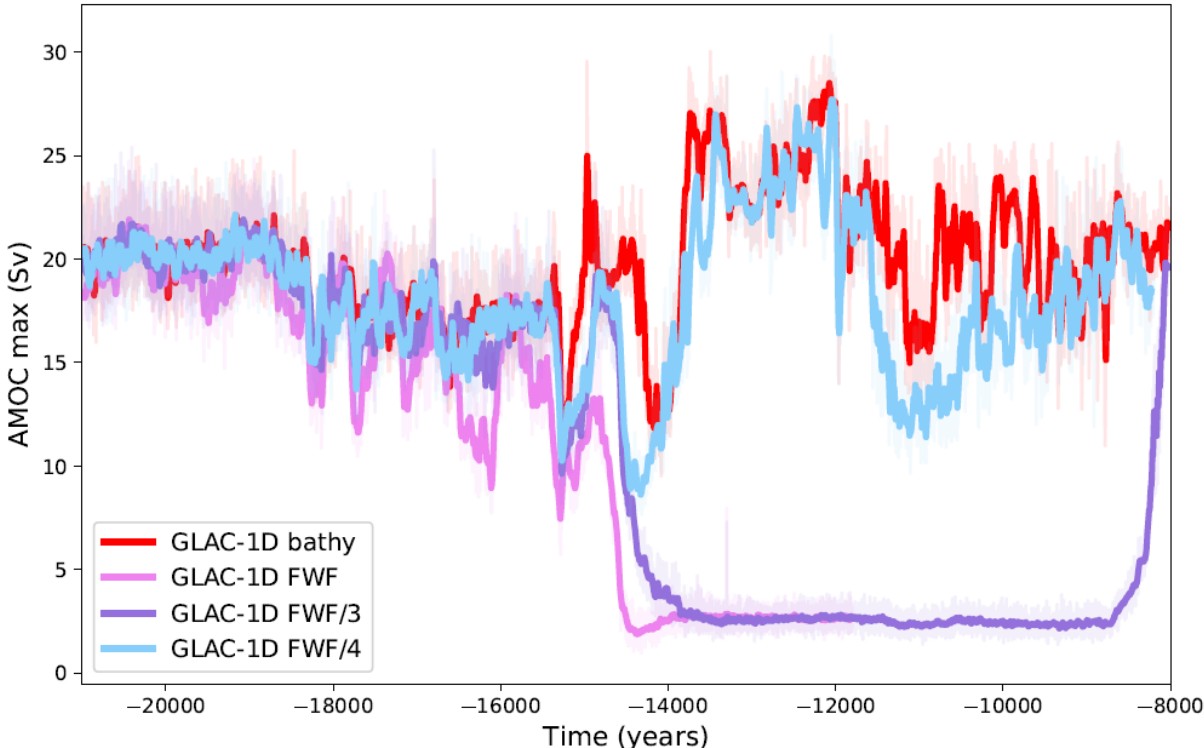

**Figure 14. Evolution of the maximum strength (Sv) of the Atlantic Meridional Oceanic Circulation (AMOC) for the simulations with fresh water fluxes.**


Because of the AMOC reduction and shutdown, less heat is brought to the North Hemisphere and the temperatures in Greenland are strongly reduced in the FWF simulation compared to the standard simulation without FWF (Figure 15b). This

is also the case when the water flux is divided by 3. With the fresh water flux divided by 4 (FWF/4) the AMOC strength is only slightly diminished, and the temperature decrease in Greenland is very limited, both in amplitude, and in time as the AMOC recovers quickly. In this case, we observe two AMOC drops (Figure 14, similar to those observed in Kapsch et al., 2022), and simultaneously two temperature drops in the Greenland temperature evolution (Figure 15b).





**Figure 15. Temperature evolution (°C) for the simulations with fresh water fluxes (a) of the global mean (b) at NGRIP (Greenland), (c) at EDC (Antarctica). The simulated results are shown as running mean over 100 years and compared to proxy data (Shakun et al., 2012; Buizert et al.; Jouzel et al., 2007).**

While the Bolling-Allerod warming in the Greenland record at ~14.5 ka cannot be simulated in GLAC-1D FWF as the fresh water flux from ice sheet melting becoming substantial only later at ~15 ka, the temperature evolution in Antarctica and in the global mean (Figure 15a and c) is in better agreement in the model compared to data, with a stalling in the temperature increase at the time of the Antarctic cold reversal from ~14.5 ka. Yet the warming that starts again at ~12 ka in the data lags in the simulations with FWF.


## 4. Discussion and conclusion

Using the iLOVECLIM model of intermediate complexity, we have evaluated the impact of different ice sheet reconstructions, separating their different effects: the change of topography (and albedo), the change of bathymetry (and land-sea mask) and the change of fresh water fluxes from the ice sheet melt.


### 4.1 Different ice sheet reconstructions

In our simulations, the ice sheets are prescribed based on two reconstructions: ICE-6G_C and GLAC-1D, following the PMIP4 protocol. We show that the two reconstructions result in differences in terms of amplitude and timing of climate changes. In particular the global mean temperature is ~0.3°C warmer with GLAC-1D compared to the simulation with ICE-6_C, from the

LGM to ~12 ka. Between 14 ka and 12 ka the evolution between the two simulations differ. Only the simulation with GLAC-1D displays a stalling in the temperature warming similar to data, albeit with a lag compared to data.

### 4.2 Bathymetry

In most past studies of deglacial simulation, the bathymetry was fixed, either at the LGM or the PI. We have shown here that

the impact of accounting for bathymetric changes is a shift towards colder values, which is relatively limited for global mean

temperature, but more important regionally. This effect is mainly due to the different albedo with and without bathymetry changes. With evolving bathymetry, some grid cells are progressively changed from continent to ocean throughout the deglaciation, increasing the space available in the Nordic seas especially. This results in a larger sea ice area with a higher albedo, which cools the climate and modifies the convection zones in the North Atlantic. As we demonstrate that bathymetry

changes can impact the local and global climate, we advise to account for them (when technically possible) in deglacial simulations.

The evolution of bathymetry depends on the ice sheet reconstructions. As the reconstructions are improved over time, the bathymetry change will also become more constrained. However, the model grid change resulting from the bathymetry change is not always straightforward when seaways become small and especially smaller than the grid cells: should a passage stay

open in the model even if it results in a larger seaway, or should it be closed, even though it is not completely closed in the reconstructions? In this study, we have chosen to keep seaways open when they were open in the reconstructions, even if it was smaller than the grid cell. Yet, it would be interesting to test the impact of closing them. This issue is related to the model resolution: with higher resolution models it is possible to keep small seaways open, while in lower resolution models the grid cells are sometimes too large.

In LGM simulations, a previous study showed that accounting for bathymetry and land-sea mask change was crucial for the carbon cycle (Lhardy et al., 2021b). A change of bathymetry, especially the one associated ocean volume change, modifies the ocean carbon storage capacity. As it is a very large carbon reservoir compared to the atmosphere, even changes of around 3%, the order of magnitude between LGM and PI volume changes, can largely impact atmospheric $CO_2$. This effect remains to be tested for the deglaciation.

In our study, the ice sheets are prescribed from reconstructions. In reality, ice sheets and climate interact. While the use of prescribed ice sheets is a limitation of this study, this topic has been addressed in Quiquet et al. (2021) with an interactive ice sheet model coupled to the iLOVECLIM model. These simulations show similar results in terms of temperature and ocean circulation changes. However, the bathymetry was fixed in such simulations. As shown here, the regional temperature can be impacted by the changes of bathymetry, which influences oceanic circulation and hence the transport of heat. In the simulations

with evolving bathymetry, the temperature is colder over Greenland and Antarctica compared to the simulation with fixed

bathymetry (Figure 7). In simulations with interactive ice sheets, this would impact the mass balance and could lead to a different evolution of ice sheets.

## 4.3 Fresh water fluxes

In iLOVECLIM, the AMOC is very sensitive to the fresh water flux, resulting in an AMOC shut down when freshwater input from ice sheet melting is included. The AMOC then remains in this collapsed state for several thousand years. The sensitivity to fresh water flux is very model dependant (Gottshalk et al., 2019), and some other models display less sensitivity. For example, with the MPI-ESM model (Kapsch et al., 2022), the same fresh water fluxes computed from the ice sheet melt from the ICE-6G_C and GLAC-1D reconstructions result in weaker AMOC but no collapse, more similar to our FWF/4 simulation.

In most past studies of deglacial simulations (apart from Kapsch et al., 2022), fresh water fluxes were prescribed and not computed from the ice sheet melting. In figure 13, we compare the fresh water fluxes used in those simulations. In most model studies, the fresh water flux is much smaller and / or with a different timing compared to the ones computed from the ICE-6G_C and GLAC-1D reconstructions. This leads to a smaller evolution of sea level equivalent than in the reconstructions (Figure 16). While in most models, fresh water fluxes are necessary to trigger large ocean circulation changes, one model has

displayed changes in ocean circulation without fresh water fluxes (Zhu et al., 2014). In this simulation, the ocean circulation change was due to the orographic change of the ice sheet only (Zhu et al., 2014). It is thus difficult for models to explain the large change of ocean circulation and the related change of temperature in the Northern and Southern Hemisphere between 18 and 15ka by freshwater fluxes. Either the ice sheet and sea level reconstructions should be revisited, the modelled AMOC sensitivity modified, or other processes explaining the AMOC change tested.




**Figure 16. Equivalent sea level evolution (m) due to the fresh water input (displayed on Figure 13).**



**Data availability**

The source data of the figures presented in the main text of the paper will be available on the Zenodo repository.

**Author contribution**

All authors have developed the model code. NB has performed the simulations. NB ran the analysis and prepared the manuscript with contributions from all co-authors.

**Acknowledgements**

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
