# Peer review of "Deglacial climate changes as forced by different ice sheet reconstructions"

_EGUsphere, 2022_

## Author Comment (AC1)

**Response to reviewer RC1**

We are thankful to the reviewer for their comments and we respond point by point in the following in blue.

Review of Deglacial climate changes as forced by ice sheet reconstructions by N. Bouttes et al.

The manuscript presents the results of the transient simulation of the last deglaciation with iLOVECLIM, following experimental designs of PMIP4. The authors present seven experiments with two different ice-sheet reconstruction datasets and different treatments in bathymetry or freshwater influx to the ocean. They show the evolutions in temperature, ocean circulations, and other climatic variables. They discuss why the simulated climate fields differ between experiments, and discuss the link between climate forcing and climate changes.

Overall, this article's experiments, results, and analysis are very good, and the manuscript is written and easy to follow. The systematic experiments in this study would help other climate modeling groups. The article is worthy of publication in the Climate of the past. Still, I would appreciate it if the authors addressed minor points as detailed below.

1) Title of the manuscript: I wonder if "different ice sheet reconstructions" or something like this phrase might be better to describe this article, as ice sheets are not the only forcing of the last deglaciation. And the evaluation of seven different experiments (two ice sheet reconstruction datasets + different boundary condition treatments) is the advantage of this article.

As suggested we have modified the title to: "Deglacial climate changes as forced by different ice sheet reconstructions".

2) L20: recommend adding "global-mean temperature".

We have done the suggested modification:
"We show that the two reconstructions yield the same warming to a first order, but with a different amplitude (global-mean temperature of 3.9°C with ICE-6G_C and 3.8°C with GLAC-1D) and evolution."

3) L25: The thing that the "link" refers to might not be clear: Link between [reconstructed freshwater fluxes and recorded AMOC] or [reconstructed freshwater fluxes and their representation]? I would recommend rephrasing this sentence.

We have rephrased as:
"This questions the causal link between reconstructed fresh water fluxes from ice sheet melt and recorded AMOC weakening."

4) L31: global mean surface temperature

done

5) L32: deglacial "surface temperature" changes?

done

6) L44-46: Reading this paragraph (L29-46), the logic of "difficult to disentangle the links" may not be clear.

We have modified this part to make it clearer:
"Ice-sheets, ocean circulation, atmospheric and oceanic temperature are thus linked, but disentangling the timing and causal links between all of their changes during the transition is difficult."

7) L79: If I understand correctly, no previous study performed simulations evaluating the relative effects as in the following paragraph. So I would recommend stating "has not been evaluated" instead of just "unclear".

done

8) L107: Please clarify the modifications. Modification in codes or just applied time-dependent forcing?

We have modified the code to be able to change the bathymetry and coastlines, as well as route the fresh water flux from melting ice sheets. We have modified the text as follows:

"To run simulations of the last deglaciation following the PMIP4 protocol, we have modified the code to be able to change the bathymetry and coastline regularly, and we have added the routing of fresh water from ice sheet melt to the ocean. We have also generated the ice sheet topography, ice sheet mask, bathymetry and land-sea mask files on the iLOVECLIM grid using the ICE-6G_C or GLAC-1D reconstructions. This is detailed in the sections below."

9) L117: Please indicate the treatment of the terrestrial biosphere in the transient simulations. Is this prescribed or forecasted in the vegetation/land surface model?

The land vegetation is computed in the model. We have added the following:
"The vegetation is dynamically computed in the model by the terrestrial biosphere model (VECODE; Brovkin et al., 1997)."

10) L218: the main text says Figure 5 is 21-8 ka, but the caption says 21-10 ka.

We have changed the text to 10 ka.

11) L234 (Figure 6): As the salinity is conserved in the simulations, the global mean ocean salinity would be simply the inverse of the ocean volume (Figure 2a), so I wonder why Figure 6 is necessary. Does the mean salinity provide peculiar information?

It is true that there is some redundancy between the two figures, but looking at the salinity allows us to check that the salinity changes have been correctly computed in the model according to the bathymetry changes. In addition, we think it will be a useful figure to be able to compare between model results as the treatment of bathymetry changes might be different and could results in different salinity values. Finally, it is also a useful information to understand carbon cycle changes in deglacial runs as this shows the treatment of ocean variables, which will be useful when carbon cycle will be looked at in the future. So, although at the current time it might not bring much extra information, we think it is worth keeping the figure of salinity evolution as it might be valuable information in future work.

12) L243: Please indicate that the vertical axis of Figure 7 uses different scales between the model and ice core data (like normalizing between LGM and Holocene).

We have added this in the caption of Figure 7:
"Note that the vertical scales for the model simulations (left) and for the measured data (right) are different."

13) L250: Please indicate the data references of the NGRIP and the EDC.

We have added the reference for the data:
"Data are from Shakun et al., 2012; Buizert et al., 2014; Jouzel et al., 2007**.**"

14) L253: I'm not convinced with "with a larger amplitude than at a global scale". I recommend writing out the number of global-mean/NGRIP/EDC temperature changes in the text following Figure 7.

As suggested we have added the numbers:
"The relative cooling in the simulations with bathymetry changes compared to the simulations with fixed bathymetry is higher locally than at a global scale, especially in Greenland. The maximum cooling is ~2°C at NGRIP (2.1°C with ICE-6G_C and 2°C with GLAC-1D) and ~1°C at EDC (1.1°C with ICE-6G_C and 0.7°C with GLAC-1D) compared to ~0.4°C globally (0.5°C with ICE-6G_C and 0.3°C with GLAC-1D)."

15) L255: One standout from Figure 7c-f is the significant difference between ice core data and two simulations because the AMOC keeps its intense mode (shown in later in Figure 11). I recommend to briefly noting here that the AMOC was mostly intense in these two simulations, and that's one reason for the significant model-data difference in Figure 7c-7f.

Following the suggestion, we have added:
"In these simulations, there is no large abrupt climate change in Greenland contrary to data, as the AMOC strength remains strong throughout the simulation in the absence of fresh water flux (see later discussion and figure 14)."

16) L257-259 (Figure 8): It is unclear which panel is explained in the main text. Does "15-14 ka onward" refer to the panel of 65-90N?

It is for all panels. Except for some specific times, the albedo is generally higher with evolving bathymetry compared to the corresponding simulation with fixed bathymetry. We have added some precision:
"As shown on Figure 8, except for a few time periods, the albedo becomes higher in all simulations with evolving bathymetry compared to the corresponding simulations with fixed bathymetry, a difference that becomes more visible from ~15-14 ka onward, at the same time as the shift in temperature observed on Figure 7."

17) Also, I would ask authors to consider adding a 2-d map of albedo at key time slice (e.g., Southern Hemisphere at 13 ka with two ice-6g_c experiments), which might help to understand.

The albedo is an important variable to understand the changes of temperature. We show below as an example the albedo map at 14ka and the difference between 13ka and ka for the simulation with ICE-6G-C and changing bathymetry. However, as we already have a large number of figures (16) and we already show evolution of albedo through time we think it is not necessary to add 2d maps of albedo.

[Figure]

[Figure]

Figure: top: albedo for the ICE-6G_C with evolving bathymetry simulation at 14 ka, bottom: albedo difference between 13ka and 14ka.

18) L262: Is it possible to indicate the typical values of the albedo of continental surface and sea ice?

The albedo varies depending on the continental surface type and snow coverage. Typical values are: Albedo of trees=0.13, albedo of grass=0.20, albedo of desert=0.33, Albedo of snow=0.8-0.85. The albedo of open sea varies between 0.06 and 0.14. The sea ice albedo depends on the ice thickness, the presence of snow, melting… It varies between 0.8 and 0.1 for very thin ice. We have added an idea of albedo values in the text:

"the continental surface whose albedo can vary from ~0.1 without snow to ~0.8 with snow is replaced by surfaces with sea ice, which has a high albedo of ~0.8 (which can decrease depending on the sea ice type, down to 0.1 for very thin ice), leading to the cold temperatures."

19) L268 (Figure 9): why there is a sharp increase in ice-6g_c evolving bathy exp at the end of the simulation?

The simulation crashes just before 8ka producing non-numerical values for some variables. We have removed the very last part of the simulation on the figure. It does not modify the rest as we have only removed the last 100 years.

20) L374: It seems FWF/3.5 in Figure 15a not used in the manuscript

Yes, we have removed it.

21) L404-411: I agree with the authors that we should account for bathymetry changes, but discussion from model-data comparison in this point would be necessary. For example, the Figure 7e experiment has a sharp warming at 13.5ka, which seems to be absent in ice core data. I wonder if this warming only in model results might be a "right" climate response but a different time period, or if it is technically still challenging points to account for realistic bathymetry/coastline changes in the model, or uncertainties in ice sheet reconstructions. Please discuss this point further.

There is indeed a response to the bathymetry change around 13.5ka that is not in agreement with data. It shows how difficult it is to correctly implement bathymetry changes and how important it is as it can trigger large changes. We have identified a grid cell that is switched from land to ocean in Antarctica at that time, and have tested to run a simulation by prescribing this grid cell to land instead of changing to ocean. The results show that as expected this suppresses the sharp albedo change that was simulated before as well as the temperature change (new Figures 7 and 8 with simulations "ICE-6G_C evolving bathy mod"), confirming the cause for the sharp simulated albedo and temperature change in Antarctica at 13.5ka. We have added this simulation and some text to discuss this and show how important accounting for bathymetry is, but also how challenging it is:

"With ICE-6G_C, Figure 7e also shows a rapid increase then decrease of temperature between 14 ka and 12 ka in Antarctica. This is due to a change in the bathymetry and sea-land mask around Antarctica, where a grid cell switches from land to ocean. Forcing this grid cell to remain as land suppresses this unrealistic response (Figure 7 simulation "ICE-6G_C evolving bathy mod")."

"However, changes of bathymetry can also trigger unrealistic changes, such as observed here in the simulations with the ICE-6G_C reconstruction at ~13.5ka around Antarctica. Using a higher resolution might help, but this shows that accounting for realistic bathymetry changes remains challenging."

22) L441-443: It may be noted that some previous studies (appears in Figure 13) used simplified areas in freshwater, compared to this study utilizing river routing.

Indeed, many studies prescribed the areas where freshwater is added to the ocean. As suggested we have added a sentence on this:

"In most past studies of deglacial simulations (apart from Kapsch et al., 2022), fresh water fluxes were prescribed and not computed from the ice sheet melting. In addition, in most previous studies the areas where fresh water flux is added are prescribed. In this study the fresh water flux is computed from the ice sheet volume change and routed towards the ocean following the topography."

23) L446-448: Please clarify the logic of this sentence, because

- The sentence discusses 18-15ka, but results from Zhu et al. (2014) exhibit AMOC reduction mainly after 15ka.

- According to Figure 16b, "smaller evolution of sea level equivalent than in the reconstructions " may not be necessarily true in terms of total sea level rise between 18-15ka.

- I guess you expect weaker AMOC at 18-15ka (based on introduction L40-41), but not sure from this paragraph.

We have rewritten this paragraph to be clearer:

"These models obtained a similar evolution of AMOC compared to proxy data and accordingly a relatively good evolution of temperature in Greenland, but the prescribed fresh water flux is in disagreement with the currently available ice sheet reconstructions. It is noteworthy that while in most models, fresh water fluxes are necessary to trigger large ocean circulation changes, one model has displayed changes in ocean circulation without fresh water fluxes (Zhu et al., 2014). In this simulation, the ocean circulation change was due to the orographic change of the ice sheet only (Zhu et al., 2014). However, the prescribed fresh water fluxes are also smaller than in the reconstruction leading to smaller sea level change compared to data. Larger fresh water fluxes similar to the data may lead to different and possibly degraded results compared to data."

24) I agree with the final sentence, "either the ice sheet and sea level reconstructions should be revisited...", but I think you may refer to previous studies (e.g., Ivanovic et al. 2018 paleoceanography), and how the present study improves the discussions on this topic.

As suggested we have added the Ivanovic et al. (2018) paper in the discussion:

"In a simulation with HadCM3, a larger sensitivity of AMOC to the fresh water flux is obtained, but only for the 19 ka to 16 ka period as the simulation was not continued after (Ivanovic et al., 2018)."

However, because the simulation with HadCM3 stops relatively early at around 16ka, meaning just before the large sea level rise and large associated fresh water fluxes, it is difficult to discuss and compare model results.

25) L24 (abstract): the phrase "This questions the links..." might be somewhat strong compared to the discussion subsection 4-3. I would recommend reconsidering the abstract sentence.

We have rephrased the sentence as follows to make it clearer:

"This questions the causal link between reconstructed fresh water fluxes from ice sheet melt and recorded AMOC weakening."

---

## Author Comment (AC2)

**Response to reviewer RC2**

We are thankful to the reviewer for their comments and we respond point by point in the following in blue.

The authors present a very interesting study with iLOVECLIM model about the last deglaciation period. The transient simulations and the sensitivity experiments unravel the relative importance of different components of interaction between cryosphere and climate. The draft is very well written and very legible and systematic, and thus recommends publication. Below are some questions, comments and some minor modifications which authors might like to consider answering and clarifying in the draft.

1) Line 115: "The updates are done abruptly, i.e. without temporal interpolation, every 500 years for the ICE-6G_C reconstruction and every 100 years for the GLAC-1D reconstruction".

Might be interesting to clarify here why such a choice has been made for the interacting time scales. Is this to match to temporal resolution of ice sheet reconstructions?

Yes, the ice sheet reconstructions are available every 500 years for ICE-6G_C (https://pmip4.lsce.ipsl.fr/doku.php/data:ice_ice6g_c) and every 100 years for GLAC-D (https://pmip4.lsce.ipsl.fr/doku.php/data:ice_glac_1d).

We have added this information:

"The difference in frequency update is due to the difference in the frequency of available ice sheet reconstructions (500 years for ICE-6G_C and every 100 years for GLAC-1D)."

2) Line 170: Can also clarify if the meltwaterflux is added uniformly across global ocean or in specific regions?

The meltwater flux is routed following the topography to the closes ocean. We have added this in the paragraph:

"With the new routing scheme, the fresh water flux from ice sheet melt is thus not added homogeneously in the ocean but routed towards the closest ocean grid cell following the topography."

3) Line 203:- Does the difference in time interval for updating ice sheet topography and albedo information in model, i.e 500 years in ICE-6G vs 100 years in GLAC-1D, can have an impact on why ICE-6G has not simulated the temperature decrease between in 14-12 kyr?

It is likely that the frequency of update has some impact, although changing from every 500 years to every 100 years with the same bathymetry reconstruction has probably less impact than using different reconstructions. To test this, we have run an additional simulation using the GLAC-1D reconstruction, and updating the bathymetry and land-sea mask every 500 years instead of 100 years. As shown on the figure below, the update frequency does have an impact, but the difference in temperature evolution compared to the simulation with the 100 year frequency is less than the one with the simulation with the other reconstruction (ICE-6G_C).

We have added this in the text (along with the new figure):

"To test the effect of the frequency update of bathymetry and land-sea mask, we have also run a simulation with GLAC-1D, updating the files every 500 years instead of 100 years."

"The update frequency for bathymetry and land-sea mask is different for the two reconstructions: 500 years for ICE-6G_C and 100 years for GLAC-1D. To test the impact of different update frequency, we have run an additional simulation with the GLAC-1D reconstruction, with an update frequency of 500 years, similar to ICE-6G_C. As shown on Figure 13, the frequency update modifies the global temperature evolution. It often results in a delayed response compared to the simulation with 100 year frequency, as the bathymetry and land-sea mask change happens later, for example at ~19 ka, ~16.5 ka, ~11 ka or ~10 ka. Yet the effect is limited, and the difference between the two simulations with the same reconstruction but different update frequency is smaller than the difference between the simulations with the two reconstructions and the same update frequency."

[Figure]

**Figure 13. Evolution of global mean temperature (°C) for the simulations with evolving bathymetry with the ICE-6G_C or GLAC-1D reconstruction. For GLAC-1D, two update frequencies (100 and 500 years) have been tested. The simulated results are shown as running mean over 100 years. Data are from Shakun et al., 2012. Note that the vertical scales for the model simulations (left) and for the measured data (right) are different.**

"In addition, we have chosen to keep the frequency of bathymetry update from the original frequency of the ice sheet reconstructions. This results in two different frequencies for the two reconstructions (500 and 100 years). Testing the impact of different update frequencies for the same reconstructions show some limited impact: a less frequent update leads to delays in the climate response as the changes take place later. Yet this effect is small compared to the change in climate from the two different reconstructions with the same update frequency."

4) Lines 234-235: "The salinity change at the LGM that is computed directly from the volume change (1.4 psu for ICE-6G_C and 1.3 psu for GLAC-1D)) is larger than the one prescribed in the standard protocol without bathymetry change (1psu)"

Does this means there must be a 0.3 Psu difference between dashed and solid lines in Figure 6 ?

No, the standard protocol refers to the PMIP4 protocol when no bathymetry change is accounted for, i.e. when the bathymetry is set to the pre-industrial (it is one option in the protocol). Here the bathymetry is set to the LGM and the salinity change computed accordingly. We have modified the sentence to make it clearer:

"The salinity change at the LGM that is computed directly from the volume change (1.4 psu for ICE-6G_C and 1.3 psu for GLAC-1D)) is larger than the one prescribed in the PMIP4 standard protocol without bathymetry change where the bathymetry is fixed to the pre-industrial one (1 psu)."

5) Line 270:- "Due to the sea level increase, the ocean surface in the high latitudes of the Northern Hemisphere increases with time"

Since this experiment does not include melt water flux input, is there significant sea level increase in this experiment due to thermal expansion of water?

The experiment does not include melt water flux directly, meaning there is no addition of fresh water in some ocean regions, but the bathymetry and land-sea masks are modified according to the ice sheet change (mostly ice retreat) so that the water coming from the melting is added to the ocean through the change of bathymetry and land-sea mask. The increase of ocean surface is due to this change of land-sea mask.

We have added more details: "Due to the sea level increase from ice sheet melt, the land-sea mask is modified, and the ocean surface in the high latitudes of the North Hemisphere increases with time"

6) It is interesting to see a consistent dip and then retrieval in albedo and seaice in southern hemisphere between 14-12k in ICE-6G_C evolving bathymetry experiments. Corresponding temperature rise is seen in time series of model temperature in EDC location as well. Did authors further investigated about processes in southern ocean resulting in such changes? It would be informative to add some inferences about this in the draft.

See Response to reviewer RC1 question number 21.

7) Line 261 "At high latitudes where it is cold enough to have sea ice, the continental surface is replaced by surfaces with sea ice, which has a higher albedo, in particular in summer, leading to the colder temperatures"

Can authors explain bit further on the difference in albedo caused by sea ice formation. Is the sea ice is replacing icesheet surfaces in higher latitudes? Is the difference because icesheet surfaces have lower albedo than sea ice? Doesn't sea ice albedo depends on how thick is the ice and how much snow is on the top etc, which all tends to reduce surface albedo compared to ice sheets?

See response to reviewer RC1 question number 18.

The albedo of ice sheet is typically 0.85. If there is no ice sheet, the continent can be covered by snow, especially in winter, whose albedo is also high, around 0.8. If the snow melts, the albedo

decreases, and when no snow remains the albedo varies between 0.1 and 0.3 depending on the vegetation. Hence if the continent is not completely recovered by snow, especially in summer, its albedo will be less than sea ice.

8) How is AMOC strength in Figure 11 is computed?

It is the maximum of the overturning circulation in the North Atlantic.

9) Why there is such large differences between sea ice formation and AMOC strength for two icesheet reconstructions with evolving bathymetry. Again, does the time interval for ice sheet and bathymetry updates to the model could produce any of the differences?

The update frequency has a limited impact, see response to question number 3.

**Minor comments/corrections:**

10) Line 199: When temperatures from both simulations have similar values,:- when temperatures from both simulations reach similar values seems more appropriate ?

done

11) line 204-205 The sentence can be rewritten as lots of comma separated statements make it difficult to understand.

As suggested we have rewritten this part:

"This can be linked to the Northern hemisphere ice sheet reconstructions. In GLAC-1D the ice volume decreases strongly until 14ka, then stays constant from 14 ka to 12 ka. With ICE-6G_C it continues to decrease, although at a lower rate."

12) Line 275:- this effect is counteracted by the increase of ocean surface at high latitudes so that the sea ice area shows a more limited evolution.

"reduction" might be more suitable than "evolution" in this sentence?

Yes, done

13) In both Figure 11,14 captions :AMOC is expanded as Atlantic Meridional Oceanic Circulation instead if Atlantic Meridioanal Overturning circulation.

This has been corrected.